# “It Is Like Compulsory to Go, but It Is still pretty Nice”: Young Children’s Views on Physical Activity Parenting and the Associated Motivational Regulation

**DOI:** 10.3390/ijerph17072315

**Published:** 2020-03-30

**Authors:** Arto Laukkanen, Arja Sääkslahti, Kaisa Aunola

**Affiliations:** 1Faculty of Sport and Health Sciences, University of Jyväskylä, 40014 Jyväskylä, Finland; arja.saakslahti@jyu.fi; 2Department of Psychology, University of Jyväskylä, 40014 Jyväskylä, Finland; kaisa.aunola@jyu.fi

**Keywords:** physical activity parenting, physical activity, parental control, children, motivation, motivational climate, qualitative research

## Abstract

Physical activity parenting (PAP) is consistently correlated with children’s physical activity (PA). Children’s perception of PAP has garnered little attention given that it mediates the relationship between PAP and child PA outcomes. This study aimed to examine 7–10-year-old children’s perspectives on PAP practices and how they relate to their motivational regulation of PA. A total of 79 children 7–10 years of age participated in 19 semi-structured focus group interviews. Through qualitative theory-guided content analysis, using frameworks of parenting dimensions and self-determination theory (SDT), we found that children’s perceptions of high responsiveness and low demandingness in PAP—according to SDT, autonomy support, involvement, and structure—were associated with satisfaction of all three psychological basic needs—autonomy, competence, and relatedness. In contrast, perceptions of high demandingness and low responsiveness in PAP, i.e., coercive control, were associated with dissatisfaction of autonomy need. However, perceptions of high demandingness and high responsiveness in PAP, specifically parental expectations and facilitation of PA, were associated with satisfaction of competence need. It seems possible to identify different types of PAP practices associated with children’s motivation for PA. Different forms of parental demandingness with differing motivational outcomes were uniquely identified from the children’s perceptions of PAP.

## 1. Introduction

Gaining an understanding of the factors enhancing physical activity (PA) is an important issue worldwide, as the prevalence of children not meeting the recommended level of PA for health is high and likely increasing [1]. Moreover, promotion of habitual PA is among the priorities of athletic development programs since the negative trends of overall PA levels likely reduce the beneficial effects of exercise and sports participation and increase the risk of sports-related injuries in children and youth [2]. Physical activity parenting (PAP) practices (i.e., concrete behavioral strategies employed by parents to influence their children’s PA) account for one of the few identified consistent correlates [3], and they have been identified as a determinant [4] of children’s PA. According to the integrated model of PAP (IMPAP), children’s perceptions of PAP functions as a mediator between parent-reported PAP and child PA [5]. Empirical research has supported this model in several ways. First, it has shown that children’s perceptions of PAP are associated with their PA [6], even more strongly than parental reports of PAP [7,8]. Second, young athletes’ perceptions of a parent-initiated motivational climate have been shown to be associated with factors such as motivational regulation [9,10,11] and continued participation in sports [12]. However, the limitation of this earlier research is that little is known about children’s perceptions of PAP and how these relate to their motivational regulation of habitual, everyday PA. Such knowledge would contribute to advancing our understanding of how PAP facilitates children’s habitual everyday PA, motivation for PA, and, ultimately, PA habit formation in childhood.

In the present study, we explored 7–10-year-old children’s perceptions of PAP and how these are associated with their motivation for PA. Children’s perceptions of PAP and the associated motivation are examined using the parenting dimensions framework [13,14] and self-determination theory (SDT) [15,16] (pp. 319–350). The parenting dimensions provided the theoretical framework for conceptualizing the children’s perceptions of parental responsiveness and demandingness. SDT provided the framework to conceptualize the children’s perceptions of their basic psychological need satisfaction and motivational regulation of PA.

### 1.1. Children’s Motivation to Physical Activity (PA)

Motivation is a psychological force that drives an individual’s intention and action. In the SDT framework, motivation is seen as being multidimensional in terms of regulatory styles, loci of causality, and corresponding processes [15]. Intrinsically and autonomously regulated motivation refers to behaviors that are seen as interesting and enjoyable. Regulation of extrinsic motivation is categorized based on the degree to which it is controlled (external or introjected) or internalized as autonomously regulated (identified or integrated). Thus, externally motivated and non-interesting behavior can be internalized (autonomously regulated) if an individual perceives it to be personally meaningful (i.e., identified) or aligning with his/her other values and goals (i.e., integrated). Amotivation refers to an absence of motivational regulation. Evidence demonstrates that autonomous forms of motivation correlate positively with PA and controlled forms of motivation and amotivation correlate negatively with PA in children and adolescents [17].

According to SDT, satisfaction of the three basic psychological needs—autonomy, competence, and relatedness—facilitates autonomous motivational regulation [15]. The need for autonomy refers to “the need of individuals to experience self-endorsement and ownership of their actions—to be self-regulating in the technical sense of that term”; the need for competence describes the need for “feeling effective in one’s interactions with the social environment—that is, experiencing one’s capacities and talents”; and the need for relatedness is defined as “both experiencing others as responsive and sensitive and being able to responsive and sensitive to them—that is, feeling connected to and involved with others and having a sense of belonging [16] (p. 86). In previous literature, need satisfaction has been shown to explain the significant variations in motivational regulation styles of PA in children aged 7–11 years [18].

### 1.2. Parental Influence on Children’s PA Motivation

The use of more PAP practices is commonly viewed as being unambiguously favorable to children’s PA [3]. However, considerable attention has been paid recently to distal levels of parenting, such as parenting dimensions, when trying to examine the associations between PAP and children’s PA outcomes. Parenting dimensions can be categorized as two orthogonal factors, as parental responsiveness and parental demandingness [14]. Responsive parenting is characterized by warmth, supportiveness, involvement, acceptance, and expressing positive feelings. Demanding parenting is characterized by limit setting, monitoring, supervision, behavioral control, and knowledge of the child’s behavior. In the previous literature, PAP practices performed with a combination of high responsiveness and low demandingness [19,20,21] have been found to be positively associated with 5–11-year-old children’s PA. Overall, the effect of PAP on a child’s PA outcomes is likely influenced by the way in which the support for PA is provided.

However, little attention has been paid in the literature to how PAP influences children’s motivation for PA. This knowledge would be crucial given the increasing independence of decision making about PA and other activities during childhood, which means greater dependence on behavioral self-regulation. According to SDT, parental influences on children’s motivational regulation for PA are based on their satisfaction with their basic psychological needs, autonomy, competence, and relatedness. Furthermore, three key elements of parenting that support need satisfaction have been proposed: Autonomy support, structure, and involvement [16] (p. 321). Taking the child’s perspective in the interaction and decision making is a core feature of autonomy-supportive parenting. From studies investigating sports participation and athletic development, it is known that parents can significantly affect their children’s internalization of autonomous motivation toward sports. For instance, 9–14-year-old athletes’ perceptions of parent-initiated mastery climate, i.e., a climate encouraging learning and trying one’s best, instead of comparing a child’s performance to norms or the performance of others (ego-oriented climate), have been shown to be positively associated with changes in global self-esteem, performance anxiety, and autonomous motivation for sports throughout the season [11]. Similar parent-initiated motivational climate influences have been found regarding 12–15-year-old boys’ intention to continue playing sports [12]. Moreover, parental influences on young athletes’ motivational processes have been found to relate to leadership styles, affective responses, and pre-performance behaviors when parents support a child’s participation and learning [10].

According to SDT, coercive control is the opposite of autonomy support. Perceptions of high parental demandingness and pressure have been shown to frustrate the satisfaction of basic psychological needs and to inhibit autonomous motivational regulation, resulting in weaker intrinsic motivation, less enjoyment, and higher boredom in young athletes [9]. It is important to note that not all parental demandingness is harmful since the way in which the demandingness is provided matters. For instance, parental pressure is known to be associated with increased levels of anxiety over time in young athletes only if they perceive the parent-initiated motivational climate as being ego-oriented, i.e., norm-oriented [22].

Structure is the second dimension of parenting in SDT; it describes the way parents organize children’s environment to facilitate competence. When parents structure the environment, they provide clear and consistent guidelines, expectations, and rules for children; they also provide children with predictable consequences for and clear feedback about their actions. Importantly, SDT proposes that the way in which parental structure is combined with autonomy-supportive versus controlling parenting makes a significant difference in the degree to which a child feels supported. Masse et al. [23] have proposed that it is necessary to combine PAP with some level of demandingness to establish the proper environment for children to be physically active. They conceptualized this type of parenting as “expectations set about PA as to when and how much PA the child should do” and as “monitoring child involvement in PA”. Through expectations and monitoring, parents structure the environment and provide necessary guidance, but they do not pressure or dominate the child. Masse et al. [23] identified other elements of parental structure, such as parental co-participation in PA with a child, facilitation of PA participation through enrolling or taking them to places where they can be active, modeling a physically active lifestyle, and restrictions of PA involvement for safety or academic concerns. To date, no studies have investigated how these PAP practices influence children’s motivation for PA.

According to SDT, parental involvement is the third key element of parenting; it is understood as the degree to which parents devote time, invest attention and resources, are caring and supportive, and show warmth and concern for being actively engaged in their children’s lives [16] (pp. 321, 327–328). Parental involvement is hypothesized to link to the satisfaction of both relatedness and competence needs; thus, it facilitates autonomous motivation in children. Research supports benefits of parental involvement, for instance, considering students’ school performance through more autonomous motivational regulation [24]. Although parental involvement has not been studied in relation to PAP as per SDT, the literature supports that PAP high in responsiveness and low in demandingness is favorably associated with the level of PA in children [19,20,21]. 

### 1.3. Children’s Perception of PAP

A social constructivist perspective assumes that children play an active role in their own socialization process and take an active role in their own behavior by interpreting the world around them [25] (pp. 4–7). However, children’s perspectives are rarely included in PAP-related research even though the preliminary evidence states doing so would enrich the understanding of parental influences on PA. Evidence states that children even younger than the age of 10 are capable of providing reliable and valid reports of others’ behaviors, including parents, on scales administered in a developmentally appropriate format [26]. Child-perceived PAP has also been shown to mediate the relationship between parent-reported PAP and the child’s PA [27]. In practice, the same PAP practice, e.g., encouragement for PA, can be expected to contribute to a child’s satisfaction or frustration of basic psychological needs depending on his/her perception of PAP.

Qualitative research states that children 6–11 years of age commonly perceive parental support for PA, and the quality of the perceived support differs based on a family’s socioeconomic status (SES) [28,29]. Children from middle-to-high SES regions have been shown to commonly perceive tangible parental support for PA, such as parental co-participation, parental modeling of PA, and opportunities for organized PA participation. Children from low SES regions perceive more verbal parental encouragement and demands as well as opportunities for participating in more unstructured activities or free play with friends [28]. Brockman et al. [30] proposed that the verbal strategies that parents of children in schools in low SES regions use to encourage them to engage in PA (via rewards and sanctions) are consistent with extrinsic motivation. In contrast, Heidelberger et al. [29] found that children from low-income regions enjoy family PA, but the lack of parental interest in child-like activities and a focus on sedentary activities decreases PA among children. While children’s motivation was not specifically examined in these studies, it can be assumed that the lack of parental interest and verbal demands hinder satisfaction of basic psychological needs—autonomy, competence, and relatedness—which inhibit autonomous regulation of motivation for PA [15]. 

Overall, a lack of knowledge of the children’s perspective on PAP and its influences on motivation for PA may partly explain why family-based PA interventions have been shown to have a small effect on children’s PA [31]. A better understanding of PAP from children’s perspectives would inform efforts to develop feasible and effective ways to promote PA and support autonomous motivation for PA through a family context. This knowledge would also advance efforts to build a solid physiological basis for athletic development [2], as well as a psychological basis given that parents have an even greater impact on an aspiring athlete’s motivation and mental well-being than coaches [11]. Therefore, this study aimed to use qualitative methods to explore 7–10-year-old children’s perceptions of PAP practices and how these are associated with (dis)satisfaction of the basic psychological needs and, thus, the motivational regulation of PA.

## 2. Materials and Methods

We followed the “Consolidated criteria for reporting qualitative research” when we reported the methods and findings of the current study [32]. Semi-structured focus group interviews were conducted for 7–10-year-old children in March 2018. Additionally, a family background questionnaire for parents was administered.

### 2.1. Personal Characteristics and Theoretical Framework

Focus groups were conducted by the first author (AL, PhD, post-doctoral researcher, 34-year-old, male, father of two young children). The focus group interview method was chosen based on the pretests (AL) of individual (*n* = 1) and group interviews (one group of two children) and the previous literature [30]. The group interview method provides a low threshold way for young children to participate in an interview with peers and to build on other’s experiences and opinions. While focus groups are used “to get a sense of some aspect of children’s collective viewpoint or lived experience”, the interactive nature of group interview may interfere with some children’s ability to find a voice [25] (p. 104). That issue was considered in various ways. First, interviewer emphasized that children are experts of their own life and each opinion is important and valuable. Overall, the reviewer aimed at creating an accepting, open, and confidential atmosphere, where unfavorable experiences were also appreciated. Prior to the interviews, the children’s classroom teachers were asked to have a drawing class around the topic of family PA. The children were asked to present their drawing in the focus groups. The purpose of this method was to give each child an easy way to start sharing own experiences and to lead the discussion to the topic [25] (p. 96). Moreover, the relationship between interviewer and the interviewees is known to influence the dynamics of the interview [25] (p. 73). The interviewer and interviewees were unknown to each other prior to the interviews. At the beginning of each interview, the interviewer stated his name, that he is a researcher, and that he is researching and interested in children’s views on family PA.

The planning, implementation, and analysis of this study was theoretically grounded on the parenting dimensions [13,14] and SDT [15,16] (pp. 319–350). The parenting dimensions provided a higher-order theoretical framework for conceptualizing the quality of parenting in terms of different combinations of responsiveness and demandingness. SDT provided a lower-order framework for theoretically conceptualizing parenting in terms of autonomy support, structure, and involvement [16] (p. 321). Children’s perceptions of PAP were interpreted by considering their association with satisfaction or dissatisfaction with their basic psychological needs, as defined in SDT. A description of the interpreted psychological need (dis)satisfaction was provided for each case. Lastly, perceptions of PAP were interpreted based on whether they supported autonomous or controlled motivational regulation or non-regulation of motivation. 

### 2.2. Participants and Study Setting

Participants were recruited purposively from socioeconomically diverse regions. Altogether, 455 informed consent forms with the background questionnaire were given to eligible children via 20 class teachers at six schools, each in a different region. A total of 134 informed consent forms (29.5%) were received, of which 94 (20.7%) included parental approval for the parents’ and children’s participation in the study, 22 (4.8%) included parental approval for participation in the background questionnaire only (data not used in the present study), 18 (4%) included parental refusal of participation in the study, and responses from the rest of the sample (70.5%) were not obtained. Out of the 94 children with parental approval to participate in the focus groups, eight children were absent, two did not participate as informed consents were received after the focus groups, and one teacher denied time for participation in the focus groups for five children. Finally, altogether 79 children participated in the focus groups.

Interviews were conducted at the participants’ schools (*n* = 6), in 19 separate groups (mean 4.05 ± 1.35 children, min 2, max 7 children), and they lasted on average 25.58 ± 5.44 min. In general, an interview was finished after identifying direct or indirect messages of tiredness or exhaustion in the children. There were 1–2 research assistants present in each of the interviews who were in charge of tracking the time and the name of speaker in the occurrence of each utterance.

The background questionnaire for parents concerned the child’s gender (girl/boy), date of birth (to calculate accurate age), participation in organized sports (yes/no), amount of PA on weekend days (0 = not at all; 1 = under 30 min/d; 2 = approx., 30–60 min/d; 3 = 1–2 h/d; and 4 = over 2 h/d), and the respondent parent’s gender, age, and educational level (1 = comprehensive school; 2 = high school/vocational school; 3 = polytechnic; 4 = university). Higher educational level was defined as a score ≥ 3.

### 2.3. Ethics and Data Collection

The study received ethical approval from the ethical committee of the University of Jyväskylä on 22 August, 2017. Parents signed informed consent for their own and their child’s participation in the study. The reviewer told all the children that their participation in the study is voluntary and they have a right to withdraw anytime without reasons or consequences and that all research data would be de-identified and stored in a secure place. 

The focus groups were audio-recorded. The focus group interview guide included non-leading, open-ended questions around the topic of PA within the family and leisure time contexts. After the free discussion of drawings, three sets of questions were asked. The first set considered positive PA experiences in the family context: “Think about leisure time and home: Are there situations in which PA feels like fun? When do these situations take place?” The general question was followed by questions relating more closely to parent-initiated autonomous and controlled motivational climates in PA: “Describe a situation when you are enthusiastic about PA, […] free to move like you want, […] directed for being physically active and play?” The second set of questions considered negative PA experiences: “Think about leisure time and home: Are there situations in which PA is not fun? When?” The following questions considered parent-initiated amotivation and controlled motivational climate in PA: “Describe a situation when you lose your enthusiasm to move, […] you are not allowed to move like you would like to, […] you are directed to do something else than being physically active? The third set of questions considered perceptions of specific phrases and actions the parents use in the PA instruction in accordance when initiating autonomous or controlled motivational climate or amotivation: “Tell me whether your parent(s) encourage, help, guide, or direct you to be physically active?” The following questions were: “Tell me what your parent does or says when it makes PA feel nice and exiting, “[…] you have to follow the orders and rules”, “[…] uninteresting and uncomfortable?” Issues that emerged were further enquired. 

A priori sample size of 60–100 was estimated based on the previous studies of this nature [28,29] and the anticipated complexity and desired level of depth for our research questions. Data saturation was monitored throughout recruitment and data collection. Following initial analysis of the 17th set of data (*n* = 73), saturation was estimated to be achieved because themes of PAP practices were found under all the second-order theoretical dimensions and there were no new themes generated. Two additional focus groups were interviewed to ensure and confirm that no new themes would emerge. The total count of coded utterances was divided, according to the lower-order parenting dimensions in the first-, second-, and third graders, respectively, as follows: Autonomy support and involvement (23, 31, 25), structure high in responsiveness and low in demandingness (9, 20, 15), structure high in responsiveness and high in demandingness (3, 4, 3), coercive control (11, 19, 19), and lack of structure (2, 4, 1).

### 2.4. Data Analysis 

The substantive content of the focus groups, rather than their conversational dynamics, was the focus of the data analysis. Thus, the audio recordings were transcribed selectively; irrelevant speech and utterances were excluded. Qualitative content analysis, using deductive theoretical and inductive data-driven approaches [33] (pp. 541–552), was employed since we wanted to verify the existing theoretical framework of parenting and enable the data to extend the current knowledge of PAP practices. One researcher (AL) closely read the transcribed data and conducted the initial analyses including notes of the points that appeared descriptive of the discussions, as well as relevant for the study focus. A preliminary coding structure was established and descriptive quotations grouped. This phase of work was reviewed by all the authors. The data were then coded by the first author (AL) using qualitative analysis software (ATLAS.ti version 7.5.18, Berlin, Germany). The coding of data was conducted on several levels simultaneously; data were organized under main themes and up to five levels of subcategories. During the process, the codes were constantly open for renaming and re-organizing, and the code definitions were developed. Special attention was paid for identifying also the differing statements and opinions because the group interviews are known to facilitate construction of collective knowledge [25] (p. 102–107). The coding process involved constantly comparing data units assigned to codes; renaming, uniting, and separating the codes; and revising code definitions. The preliminary coding structure was revised (AL) as new relevant concepts and meanings were identified. The data were deductively organized within the parenting dimensions—responsiveness and demandingness—and their lower-order dimensions—autonomy support, involvement, (lack of) structure, and coercive control. The third-level subcategories, PAP practices, were partly theory-driven and derived from the questions used in the focus groups (e.g., describe a situation when you are free to move like you want → supporting children’s independence in physically active play and mobility), and partly data-driven, naming concepts rooted in participant responses (e.g., describe a situation when you lose your enthusiasm to move → “I don’t terribly like it when all the other family members ski much faster so then I must always ski at full speed” → co-participation in PA with family members). The process-outcome analysis, a subtype of abductive analysis, was applied to interpret the motivational outcomes associated with the PAP perceptions [33] (pp. 560–566). Thus, the fourth and fifth levels of the subcategories—psychological need and description of the need (dis)satisfaction, and supported motivational regulation style, respectively—were partly theory-driven and interpreted based on the questions used (e.g., “Describe a situation when you are free to move?” → high responsiveness, low demandingness → autonomy support → satisfaction of need for autonomy because of self-determined PA → autonomous regulation) and partly data-driven, naming concepts rooted in participant responses (e.g., “Usually, if others would like to go swimming and I don’t, so then I have to go anyway, too.” → low responsiveness, high demandingness → coercive control → dissatisfaction of need for autonomy because of involuntariness → controlled regulation). The results of the coding were then discussed and overarching themes identified.

Selected background variables were categorized to provide a PA-related framework for the children’s quotes. In detail, gender was labeled as a girl or boy and PA level as inactive (<60 min of PA per weekend day; 17.7%), active (1–2 h of PA per weekend day; 35.4%), or highly active (>2 h of PA per weekend day; 46.8%). The cut-off points between physically inactive and active children and between physically active and highly active children were determined on the basis of global recommendations of at least 60 min of moderate-to-vigorous PA per day for health [34]. Descriptive statistics were used for the background characteristics of the study sample using the statistical software package IBM SPSS, version 24.0 (SPSS Finland, Espoo, Finland) (Table 1).

## 3. Results

### 3.1. PAP Associated with Autonomous Motivational Regulation

#### 3.1.1. High Responsiveness and Low Demandingness

Autonomy support. Children’s perceptions of self-determination and independence in PA and mobility were consistently associated with reflections of joy and positive experiences; thus, they were interpreted as supporting satisfaction of autonomy need and autonomous motivational regulation for PA (Table 2). These experiences were based on the trust between child and parent: “... and parents have not arrived home yet. I can decide whether I would like to go ice skating, skiing, walking, or playing outdoors.” (boy, inactive). Children’s perceptions of parental encouragement and praise reflecting the warmth and reasonable expectations represented another set of experiences interpreted as supporting satisfaction of autonomy need and thus autonomous motivational regulation. Typically, perceptions of child-based encouragement and praise related to success in sports competitions and successful performance in sports. Encouragement and praise were provided privately between the parent and the child (“when we are downhill skiing with my mum, she always says that it went very well”; boy, highly active), as well as publicly in games (“fourth place is the most common one I get, but they still praise me so that it was still quite a good placing”; boy, highly active). Parental encouragement and praise were reported only by a few female interviewees, and they were highly task-oriented (“they encourage me if something has gone wrong with me. Mum and dad encourage like, ‘It’s a little mistake, without mistakes one cannot learn’”; girl, inactive).

Involvement. Some of the PAP experiences originated in situations in which parents listened to the children’s interests and feelings of PA (“mum asked what we could do, so then we went for hiking with my little brother”; girl, highly active) or respected their lack of motivation for participating in sports training (“if I would have ice hockey training, I don’t need to go there but I can keep playing on the free ice area because there one can move a lot more than in the training”; boy, highly active) (Table 2). These perceptions were seen to support satisfaction of need for relatedness (therefore autonomous regulation of motivation) because parents were seen to be unconditionally supportive and investing their attention in the child’s thoughts and feelings.

Structure. Co-participation in physical activities with parents was frequently reflected in the children’s stories and consistently associated with satisfaction of the need for relatedness (Table 2). Typically, co-participation related to spending time together outdoors, as well as moving around and doing home tasks together with parents, for example: “we usually go for a walk with dogs; our family goes for forest hikes” (boy, highly active). Co-participation was also commonly related to exercising or frolicking together with father outdoors, for example: “I have a competition with dad and little sister. […] I guess it’s running” (girl, highly active). Children generally talked about many opportunities to participate in organized sports, and they mainly reflected satisfaction of competence need. Children who were highly physically active described participation in organized sports as being associated with feelings of competence, enjoyment, and satisfaction, for example: “Pretty often in the football training, they [coaches] say to my teammates: ‘Take a look at how he (name of the boy) is performing’” (boy, highly active). However, physically inactive children were neutral in their reflections about organized sports. Typically, they expressed that they were simply participating in sports because of a sense of responsibility. Importantly, these expressions reflected satisfaction of the autonomy need due to parental support for quitting or changing hobbies. For instance: “I do football twice a week in the summer, plus tournaments. […] I have quit many hobbies ‘cause I have not liked doing them anymore. I have quit those ‘cause I have no longer liked those.” (boy, inactive). Thus, parental support likely supported internalization of an identified style of autonomous motivational regulation.

#### 3.1.2. High Responsiveness and High Demandingness

Structure. Children’s perceptions of demanding, goal-directed, and strict parental expectations toward PA and facilitation of PA were associated with positive affective outcomes, and they referred to the satisfaction of their need for competence and autonomous regulation on two specific conditions (Table 2). Expectations and facilitation were perceived as positive and competence-supportive if they, first, considered the children’s interests: “after all, although dad says, ‘Go there,’ usually when I am there and I go there to ice skate, I am happy that I have gone there” (boy, highly active). The second way considered parental co-participation in PA, for instance: “Usually, when mum says that ‘Let’s go for (a place to do cross-country skiing)’, you cannot argue with her so it is like compulsory to go but it’s still pretty nice” (girl, active), and “In most cases, dad, if we are swimming in his workplace, says that I must swim from one end of the pool to another end, although it is still a bit fun” (girl, inactive).

### 3.2. PAP Associated with Controlled Motivational Regulation

#### Low Responsiveness and High Demandingness

Coercive control. Experiences reflecting low responsiveness and high demandingness, i.e., coercive control, in parenting were usually associated with dissatisfaction of the need for autonomy; thus, they contributed to controlled regulation of motivation (Table 2). First, children perceived parents as forcing them to go outdoors because they spent too much time indoors and as a way of regulating their screen time. For example, “They say every day, ‘(name of the boy), now get away from the computer, television, or cell phone’. Then I need to take (dog’s name) for a walk” (boy, highly active). Forceful parental assertions also related to keeping up with participation in organized PA or sports, for example: “If I go to play a sport, they are like, ‘Now go there,’ just for getting me to move” (girl, active). Co-participation in PA with family members was sometimes perceived controlling, as well: “I don’t terribly like it when all the other family members ski much faster so then I must always ski at full speed” (boy, active).

Another set of experiences of coercive parental control related to overt and public parental encouragement and praise (Table 2). These PAP behaviors took place typically in the children’s games or competitions, and caused nervousness (“If my parent praises me too much, it starts a bit to make me nervous whether I am now doing it right”; girl, active), or feeling of shame (“they are always shouting at me like, ‘Good, good!’. But then it starts to shame me; it is bad”; boy, highly active). However, one boy found both his parents’ and coach’s overt and public encouragement and praises fully acceptable (boy, highly active). The third-way encouragement and praise were perceived as controlling related to oppressive steering of performance, for instance: “When I practice freestyle skiing…so mum tries to teach me. I do not like it when mum is always teaching and teaching. I cannot cope with it. I want to always practice by myself” (girl, highly active).

Lastly, children expressed the parents’ decision to prioritize other interests over their PA interest as a way to prevent or limit their PA opportunities (Table 2). For instance, a third-grader stated: “If we are leaving somewhere, then you don’t have time to go out. Then you need to be totally clean when we are leaving somewhere” (girl, active). Parents also repressed satisfaction of the autonomy need by requiring physically stressful home duties, which were seen as compulsory: “When mum and dad have told me to go shopping, I would have liked to go out for a change” (girl, inactive). 

Involvement. Interestingly, some parental statements illustrating high responsiveness and low demandingness were interpreted to suppress need for autonomy. Namely, two girls found parental comfort as intrusive and controlling the children’s feelings when it had taken place after a disappointment in sports. In one of the girls’ words: “If we have had a very tight game and have lost that. When my parent says, ‘Good, it went great!’ it doesn’t feel nice” (girl, active).

### 3.3. PAP with No Associated Motivational Regulation

#### Low Responsiveness and Low Demandingness

Lack of structure. Some children’s perceptions of PAP considered parenting low both in responsiveness and demandingness (Table 2). These statements related to parent’s careless and, in some cases, endlessly flexible attitude for regulating their children’s screen use. This kind of PAP was labeled as a lack of parental structure because it encouraged sedentary behavior and, consequently, inhibited feeling of competence in PA and contributed to non-regulation of motivation for PA. Children expressed lack of structure for PA in two ways, through lax screen parenting (“Five hours of war game … then, I am told that I can play only two more hours. Then I go to play with a friend at times”; boy, active) and parents’ own screen use alongside their children’s screen use (“Mum is like knitting or watching Grey’s Anatomy and dad is watching on his phone”; girl, inactive).

## 4. Discussion

The present study aimed to provide a theory-guided understanding of how 7–10-year-old children perceive PAP and how these perceptions relate to their motivational regulation of PA. Based on children’s perceptions categorized within the theoretical framework of parenting dimensions [13,14] and under the key elements of parenting according to SDT [16] (pp. 319–350), we found that it was possible to identify PAP practices that satisfy or dissatisfy children’s basic psychological needs; thus, they can be expected to contribute to or impede autonomous motivational regulation [15]. Virtually all of the previous research considering parental influences on children’s motivation for PA has focused on athlete development [11]. While none would deny the benefits of sports participation, knowledge of ways to promote habitual PA beginning at an early age is a public health priority [1]; it has also been identified as essential in the sports domain [2]. This study contributes to the literature by presenting a theory-framed perspective of children’s perceptions of PAP practices, which has the potential to enhance the present PA behavior and build a sustainable foundation for habitual PA through reinforcement of autonomous motivational regulation.

This research study adds to the current understanding, especially concerning PAP practices provided in a highly responsive and highly demanding manner. While it has been reported that parenting that combines high responsiveness and high demandingness is most likely to be associated with children’s PA [35] and favorable behavior and development [36] (pp. 11–34), only a combination of low demandingness and high responsiveness has been shown to moderate the association between PAP practices and children’s PA [19,20,21]. Thus, the latter finding conflicts with the previous evidence proposing as the ideal balance of parental responsiveness and demandingness. It has been suggested that this conflict might indicate that PAP behaviors differ, e.g., in comparison to those of the well-established children’s diet and parenting associations [19]. However, different conceptualizations of parental control have been proposed to explain the inconsistent findings in the literature [35,36]. Baumrind [36] (pp. 11–34) has proposed that a “definitional drift” has led to development and use of parenting instruments failing to distinct parental demandingness with favorable and unfavorable influences on a child. Some research has conceptualized demandingness as one-dimensional [10] rather than as two orthogonal factors; this has led to a narrow and incorrect perspective on the meaning of demandingness. The unique finding of the present study suggests that, based on 7–10-year-old children’s perceptions of PAP, it is possible to identify different types of demandingness (structure-related and coercive) and these are associated with distinct motivational outcomes. Masse et al. [23] proposed that expectations and monitoring would encompass high demandingness and responsiveness in PAP, providing a necessary structure for a child without being coercive. In addition to parental expectations, the present study suggests that children perceive that facilitation of PA, even by persuasion and power-assertive statements, is acceptable parental demandingness in PAP, especially when combined with familiarity with the child’s PA interests or parental co-participation in PA. According to SDT [16] (p. 327) and the findings of the present study, parental expectations and facilitation of PA would be expected to satisfy children’s psychological need for competence, especially when provided with consideration of the child’s PA interests or co-participation in PA.

The findings of the present study provide support for the previous studies showing that PAP practices are associated with higher levels of children’s PA, when combined with high responsiveness and low demandingness [19], and parental verbal demands and pressure to contribute to children’s controlled motivational regulation, especially when inhibiting satisfaction of the autonomy need [9,10,28]. Even though participation in organized PA or sports were primarily associated with positive perceptions of PAP, a significant amount of the perceived coercive, forceful, and disruptive parental behaviors is related to sports participation, as manifested in involuntariness, shame, embarrassment, and even humiliation. Additionally, some children perceived coercive control in relation to co-participation in PA or sports with family members; this finding is unique in the field and was made possible by the inclusion of the inductive approach. It can be speculated that the twofold character of PAP perceptions, especially about sports participation, may relate to differences in the children’s overall PA levels. Physically inactive children typically did not reflect any positive experiences of participation in organized sports, while the reflections of physically active and highly active children were mainly positive. This finding supports the acknowledged need for promotion of overall PA among aspiring athletes [2].

The present study highlighted the complexity of parenting concerning children’s screen use. The children’s perceptions of coercive parenting practices and the need to find a balance between their screen time and physically active outdoor time frustrated their need for autonomy in PA. Parents’ endlessly flexible attitude for screen use meant a lack of structure for PA; thus, it frustrated the need for competence and the development of motivational regulation for PA. Parents may perceive coercively provided rules toward screen use as leading to favorable behavioral outcomes in the short-term, but the frustration of psychological needs and, therefore, the development of controlled motivational regulation in children, does not likely lead to increased PA in the long term. Thus, more research is warranted, preferably longitudinal studies that include the children’s perspectives, to determine how parents can facilitate children’s autonomous motivational regulation of PA while simultaneously maintaining appropriate rules for screen use.

When interpreting these findings, it should be noted that numerous factors may influence children’s perceptions of parental support for PA. For instance, perceptions of parenting may be influenced by children’s individual goals for sports, timing, and the context of the parenting experiences, and the general quality of the parent–child relationship, e.g., the gender of both the child and the parent. It is also important to note that when cultural values and realities vary, parents’ objectives and the specific family processes that are most effective for accomplishing those objectives may also vary [36] (p. 28). It should be also noted that the sample was overrepresented by physically active or highly active children (82.2%) and children with highly educated parents (72.5%). Therefore, the findings of the present study need to be generalized with caution and in consideration of cultural values and attitudes. One limitation relates to the fact that only one author (AL) coded the data and reliability of the analyses was not ensured in this respect. However, the preliminary coding structure and descriptive quotations were critically reviewed by all the three authors that may improve the reliability of the analysis. Additionally, the relatively strong deductive approach in the data analysis undoubtedly influenced the findings of the study. Further qualitative work is encouraged as it has potential to reveal new subthemes of PAP, an outcome which is valuable itself and would enable, e.g., development and refinement of quantitative measures reflecting (dis)satisfaction of children’s psychological basic needs in PAP-context. A stronger inductive approach would probably reveal more nuances considering, for instance, perceptions of co-participation and would, therefore, advance understanding of the influences of specific PAP practices on children’s motivation for PA. Overall, more research on children’s perspectives on PAP is warranted as it is an understudied field and likely mediates parental influences on children’s PA outcomes [27]. 

## 5. Conclusions

The findings suggest that young children’s perceptions of PAP are associated with their motivation for PA. Ideally, parents should provide a significant amount of structure to support PA and sports participation. They should also address the need for autonomy and warmth in the interaction when considering the children’s values and interests and when providing purposeful guidance and instruction. Coercive control should be avoided as it is consistently associated with unfavorable motivational PA outcomes in children. Overall, it is important to consider children’s perceptions when promoting their PA. Consideration of the children’s perceptions of PAP could enhance development of tailored PA intervention programs, e.g., through goal setting in PA counseling with parents focusing more on autonomy-supportive and less controlling PAP practices.

## Figures and Tables

**Table 1 ijerph-17-02315-t001:** Descriptive statistics for children participating in the focus groups and their parents.

Variable	Units of Analysis	First-Graders	Second-Graders	Third-Graders	All
		Mean ± SD	Mean (SD)	Mean (SD)	Mean (SD)
Child characteristics					
*N*	Count	23	31	25	79
Gender					
Girl	Count	9 (45%)	18 (58.1%)	11 (44%)	38 (48.1%)
Boy	Count	14 (55%)	13 (41.9%)	14 (46%)	41 (51.9%)
Age	Years	7.8 ± 0.3	8.7 ± 0.3	9.7 ± 0.3	8.7 ± 0.8
Participation in sports (%)	Yes / no	73.9	67.7	80	73.4
PA on weekend days	Possible range 1–5	4.2 ± 0.9	4.24 ± 0.7	4.44 ± 0.8	4.3 ± 0.8
Respondent-parent characteristics					
Gender					
Female	Count	21 (80.6%)	23 (79.3%)	19 (76%)	63 (81.8%)
Male	Count	2 (19.4%)	6 (20.7%)	6 (24%)	14 (18.2%)
Age	Years	37.9 ± 5.3	38.5 ± 6.3	40.5 ± 4.9	38.99 ± 5.6
Higher educational level (%)	Yes / no	73.91	64.52	80	72.15

Note. PA = physical activity.

**Table 2 ijerph-17-02315-t002:** Children’s perceptions of physical activity parenting and the associated motivational outcomes.

Theoretical Framework of Parenting		Children’s Perceptions	Associated Motivational Outcomes	
Higher-Order Dimension	Lower-Order Dimension	Physical Activity Parenting Practice	Psychological Need and Description of the Need (Dis)Satisfaction	Supported Motivational Regulation Style for PA
High responsiveness, low demandingness	Autonomy support	Supporting children’s independence in physically active play and mobility	Autonomy: support and trust for self-determined PA	Autonomous
		Encouraging and praising for PA	Autonomy: unconditional approval and encouragement for PA	Autonomous
	Involvement	Listening to and asking about children’s PA interests and respecting their lack of motivation for PA	Relatedness: receiving attention and unconditional support	Autonomous
		Comforting a competition-oriented child after disappointment	Autonomy: feeling parental comfort as intrusive and controlling the feelings of sadness	Controlled
	Structure	Co-participation in PA with children	Relatedness: perceptions of togetherness with family members in PA or sports	Autonomous
		Providing opportunities for participation in organized sports	Competence: experiences of proficiency or performing sports just for duty	Autonomous
			Autonomy: support for quitting or changing hobby	Autonomous
High responsiveness, high demandingness	Structure	Providing expectations and facilitating physical activity via co-participation in PA and consideration of children’s interests	Competence: persuaded for being physically active and pleased for it	Autonomous
Low responsiveness, high demandingness	Coercive control	Forceful assertion and pressure for performing PA or sports	Autonomy: involuntary PA or sports	Controlled
		Overt, public, and competition-oriented encouragement and praise	Autonomy: interrupted, embarrassed, or shamed in PA	Controlled
		Setting other interests ahead of children’s PA interests	Autonomy: forbidden PA or involuntary PA	Controlled
Low responsiveness, low demandingness	Lack of structure	Lax screen parenting and own screen use alongside children’s screen use	Competence: lack of structure for PA	Non-regulation

Note. PA = physical activity.

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
