# Peer review of "“It Is Like Compulsory to Go, but It Is still pretty Nice”: Young Children’s Views on Physical Activity Parenting and the Associated Motivational Regulation"

_ijerph, 2020, doi:10.3390/ijerph17072315_

Round 1
Reviewer 1 Report
Firstly, let me say how much I enjoyed reading this manuscript. There are just two very minor language issues:
In the Abstract; take out the word 'is' in the sentence It (is) seems possible to identify . . .
Page 5 Line 199 Replace the word 'of' with 'in'.
Author Response
We value the time you (editors and reviewers) have taken to review our manuscript, and thank you for the insightful comments that have helped us to improve our paper. All the changes are made by using the tracked changes -function in Word. Please find below the point-by-point responses to your comments with references to page and line numbers in the revised manuscript.
#1 Reviewer
Firstly, let me say how much I enjoyed reading this manuscript. There are just two very minor language issues:
COMMENT: In the Abstract; take out the word 'is' in the sentence It (is) seems possible to identify . .
RESONSE: Thanks for the comment. Text was modified according to the suggestion (page 1 line 23).
COMMENT: Page 5 Line 199 Replace the word 'of' with 'in'.
RESPONSE: Thanks for the comment. Text was modified according to the suggestion (page 5 line 199).
Reviewer 2 Report
Thank you for the chance of reading your article. The detail of your work is generally very good and thorough. I feel that this manuscript is a good contribution for the literature. Specifically, the clarity you have provided across all of the sections. Honestly, any comment or discussion on my part would be unnecessary given the quality of this manuscript. Congratulations.
Author Response
We value the time you (editors and reviewers) have taken to review our manuscript, and thank you for the insightful comments that have helped us to improve our paper. All the changes are made by using the tracked changes -function in Word. Please find below the point-by-point responses to your comments with references to page and line numbers in the revised manuscript.
#2 Reviewer
Thank you for the chance of reading your article. The detail of your work is generally very good and thorough. I feel that this manuscript is a good contribution for the literature. Specifically, the clarity you have provided across all of the sections. Honestly, any comment or discussion on my part would be unnecessary given the quality of this manuscript. Congratulations.
RESPONSE: Thanks for these encouraging words.
Reviewer 3 Report
The present manuscript presents a qualitative study evaluating children’s perceptions of physical activity parenting (PAP) and how this relates to their motivational regulation for physical activity. Overall the manuscript is well-written and provides some novel insights into an important topic. I think this paper has the potential to make an important contribution to the literature but I have a few questions/ comments that I believe will further strengthen this manuscript.
- Please add further discussion of why a qualitative approach was implemented rather than a quantitative or mixed methods approach. The authors reference articles indicating that children can reliably report on parenting practices and reference other articles that have measured child-reported PAP (page 4, lines 146-150). Furthermore, previous researchers have evaluated self-determined motivation for physical activity in young children using validated scales (see Sebire et al 2013, IJBNPA). The reader would benefit from a clear explanation of what is to be gained from implementing a qualitative approach that might not be captured in a purely quantitative approach (e.g., richness of themes).
- The authors indicate that both an inductive and deductive approach was used, but the results focus primarily on the deductive theory-driven themes. Please clarify what if any themes were identified using the inductive approach.
- Was the first author (AL) the only one to code the data? If so, I strongly recommend having at least one other individual code the data so that reliability can be established.
- How was data saturation defined (page 6, line 257)? Relatedly, it would be helpful if the authors added in information about the frequency at which the different themes were reported across the 19 sessions. This would help the reader to get a sense of how representative the themes were across children, or if only a handful of children focused on certain themes.
- How do the authors think that having primarily active or highly active children (82.2%) influenced the results? Please consider including this as a limitation of this study.
- I thought it was interesting that parental encouragement and praise were reported by only a few female participants (page 8, lines 321-323). The authors might consider discussing this further as a potential underlying mechanism for lower physical activity engagement in girls than boys.
- The authors may not have a large enough sample to comment on this, but if possible, I was curious if they observed any differences in themes by gender or age group?
- The authors acknowledge that cultural values play a central role in parenting, which may limit the generalizability of these findings (page 12, lines 479-482). I suggest that the authors also acknowledge that this appears to be a relatively high SES sample (based on the higher education demographic information in Table 1), which may also limit generalizability.
- The authors propose that theory-guided quantitative tools for measuring young children’s perceptions of PAP need to be developed to examine the role of perceptions of PAP for children’s physical activity (page 13, lines 486-488). In what ways could the results from this manuscript inform the development of such tools? Relatedly, how could the present findings inform the development of intervention programs for promoting positive PAP, parental responsiveness, and autonomy support?
Author Response
We value the time you (editors and reviewers) have taken to review our manuscript, and thank you for the insightful comments that have helped us to improve our paper. All the changes are made by using the tracked changes -function in Word. Please find below the point-by-point responses to your comments with references to page and line numbers in the revised manuscript.
#3 Reviewer
The present manuscript presents a qualitative study evaluating children’s perceptions of physical activity parenting (PAP) and how this relates to their motivational regulation for physical activity. Overall the manuscript is well-written and provides some novel insights into an important topic. I think this paper has the potential to make an important contribution to the literature but I have a few questions/ comments that I believe will further strengthen this manuscript.
COMMENT: Please add further discussion of why a qualitative approach was implemented rather than a quantitative or mixed methods approach. The authors reference articles indicating that children can reliably report on parenting practices and reference other articles that have measured child-reported PAP (page 4, lines 146-150). Furthermore, previous researchers have evaluated self-determined motivation for physical activity in young children using validated scales (see Sebire et al 2013, IJBNPA). The reader would benefit from a clear explanation of what is to be gained from implementing a qualitative approach that might not be captured in a purely quantitative approach (e.g., richness of themes).
RESPONSE: Thanks for the helpful comment. It is true that although use of a qualitative approach is indirectly justified in the introduction (e.g. page 4 lines 144-146; page 4 lines 168-170), a clear justification and guidance for further qualitative research is weak. Therefore, we modified the last paragraph of the discussion section in the following way (page 13; lines 505-511): “A further qualitative work is encouraged as it has potential to reveal new subthemes of PAP, an outcome which is valuable itself and which would enable, e.g., development and refinement of quantitative measures reflecting (dis)satisfaction of children’s psychological basic needs in PAP-context. A stronger inductive approach would probably reveal more nuances considering, for instance perceptions of co-participation and would therefore advance understanding on the influences of specific PAP practices on children’s motivation for PA.”
COMMENT: The authors indicate that both an inductive and deductive approach was used, but the results focus primarily on the deductive theory-driven themes. Please clarify what if any themes were identified using the inductive approach.
RESPONSE: We believe it is clear that the themes of the first two categories, named as “higher and lower-order dimensions” are theory-driven. However, it is probably much more difficult to make a clear distinction between deductive and inductive approaches when it comes to the third level themes, namely the children’s physical activity parenting practice perceptions. The process of analysis was described in the methods as follows (see original peer-reviewed paper, page 6 lines 279-283): “…partly derived from the questions used in the focus groups (e.g., situations when parental actions make PA feel like taking their advice à forceful assertion for performing PA), and partly data-driven, naming concepts rooted in participant responses (e.g., situations when your enthusiasm for PA increased à listening and asking about children’s PA interests).” We realized that the above mentioned examples and the description of process were not the best ones, and we thus modified them to better illustrate the deductive vs inductive approaches used, as follows (page 7 lines 290-296): “…partly theory-driven and derived from the questions used in the focus groups (e.g., describe a situation when you are free to move like you want à supporting children’s independence in physically active play and mobility), and partly data-driven, naming concepts rooted in participant responses (e.g., describe a situation when you lose your enthusiasm to move à “I don’t terribly like it when all the other family members ski much faster so then I must always ski at full speed” à coparticipation in PA with family members).”
Additionally, we acknowledge that we could make it clearer which of the findings were derived inductively. We think that the following two subthemes are the ones most clearly derived from the data-driven approach: First, “coparticipation in PA with family members” which was perceived as controlling (page 11 line 388; page 12 line 459). Second, “comforting a competition-oriented child after disappointment” which was perceived as frustrating need for autonomy and thus contributing to controlled motivational regulation (page 11 lines 407-411). These both findings sound natural and logical but to the authors’ knowledge these have not been reported ever before in the PAP-literature. We wanted to highlight the meaning of inductive approach and took the former case as an example. Thus, we modified discussion of the results in the following way (page 12 lines 470-472): “Additionally, some children perceived coercive control in relation to co-participation in PA or sports with family members; this finding is unique in the field and was made possible by inclusion of the inductive approach”. As written earlier, the possibilities of inductive approach are discussed in more detail also in the last paragraph of the discussion section (page 13 lines 508-511).
COMMENT: Was the first author (AL) the only one to code the data? If so, I strongly recommend having at least one other individual code the data so that reliability can be established.
RESPONSE: Yes, only one author coded the data and we admit that this is a limitation and it should be acknowledged. We added that issue in the limitations section as follows (page 13 lines 498-501): “One limitation relates to the fact that only one author (AL) coded the data and reliability of the analyses was not ensured in this respect. However, the preliminary coding structure and descriptive quotations were critically reviewed by all the three authors which may improve reliability of the analysis.”
COMMENT: How was data saturation defined (page 6, line 257)? Relatedly, it would be helpful if the authors added in information about the frequency at which the different themes were reported across the 19 sessions. This would help the reader to get a sense of how representative the themes were across children, or if only a handful of children focused on certain themes.
RESPONSE: Thanks for the comment. We have clarified a priori sample size estimation (because it relates to saturation estimation), definition of data saturation and description of the saturation estimation in the methods section, as follows (page 6 lines 257-268): “A priori sample size of 60-100 was estimated based on the previous studies of this nature [28],[29] and the anticipated complexity and desired level of depth for our research questions. Data saturation was monitored throughout recruitment and data collection. Following initial analysis of the 17th set of data (n = 73), saturation was estimated to be achieved because themes of PAP practices were found under all the second-order theoretical dimensions and there were no new themes generated. Two additional focus groups were interviewed to ensure and confirm that no new themes would emerge. The total count of coded utterances was divided, according to the lower-order parenting dimensions in the first-, second-, and third graders, respectively, as follows: autonomy support and involvement (23, 31, 25), structure high in responsiveness and low in demandingness (9, 20, 15), structure high in responsiveness and high in demandingness (3, 4, 3), coercive control (11, 19, 19), and lack of structure (2, 4, 1).”
COMMENT: How do the authors think that having primarily active or highly active children (82.2%) influenced the results? Please consider including this as a limitation of this study.
RESPONSE: Thanks for the comment. It is true that the sample is likely overrepresented by physically active children although parental reports are widely known to overestimate the children’s physical activity level compared to objective measures. Additionally, in the later comment you also notice generalizability issue regarding the high percentage of children with highly educated parents. We added acknowledgement of these both generalizability issues in the limitations paragraph as follows (page 13 lines 495-496): “It should be also noted that the sample was overrepresented by physically active or highly active children (82.2 %) and children with highly educated parents (72.5 %).” The following sentence (original, not changed) deals with the possible consequences of these additional limitations: “Therefore, the findings of the present study need to be generalized with caution, and in consideration of cultural values and attitudes.”
COMMENT: I thought it was interesting that parental encouragement and praise were reported by only a few female participants (page 8, lines 321-323). The authors might consider discussing this further as a potential underlying mechanism for lower physical activity engagement in girls than boys.
RESPONSE: Thanks for the interesting notion. It is true that this difference might relate to the widely acknowledged physical activity differences between genders. However, this study did not aim to examine differences in physical activity levels or differences between genders, and we have had to focus discussion for the primary aims of the study (because of limited space). However, the small number of female participants acknowledging parental encouragement and praise may relate to group dynamics of the focus groups: girls may not be as eager to express the praises and encouragement as boys and boys may take the voice for this part. It would be thus worth of further investigation to examine whether the quality (e.g. task vs norm orientation) of parental encouragement and praise differ between genders in general. Examination of the gender question would be one opportunity in the future studies, especially in quantitative studies with adequate sample sizes. Again, because these things were not primarily aimed to be investigated in the present study, we decided not to include more analyses or discussion for this part.
COMMENT: The authors may not have a large enough sample to comment on this, but if possible, I was curious if they observed any differences in themes by gender or age group?
RESPONSE: Thanks for the comment. Statistical comparisons within themes would not be meaningful due to the small sample sizes, just like you thought. Just for your knowledge, we have made gender comparisons considering the total frequencies of coded utterances and no significant differences between genders were found. As well, we have made comparisons between the total coded utterances within the second-order dimensions and no significant differences were found although frequencies are seemingly differing. This results is probably due to the limited sample size. Overall, these analyses are not shown because gender comparisons, or other statistical tests were excluded due to challenges with the paper length. However, frequencies of coded utterances under second-order dimension themes at different grade levels are now visible due to the extension considering saturation estimation (see page 6 lines 264-268).
COMMENT: The authors acknowledge that cultural values play a central role in parenting, which may limit the generalizability of these findings (page 12, lines 479-482). I suggest that the authors also acknowledge that this appears to be a relatively high SES sample (based on the higher education demographic information in Table 1), which may also limit generalizability.
RESPONSE: Thanks for the comment. We have answered to this comment already above. In brief, this limitation is added to the discussion section.
COMMENT: The authors propose that theory-guided quantitative tools for measuring young children’s perceptions of PAP need to be developed to examine the role of perceptions of PAP for children’s physical activity (page 13, lines 486-488). In what ways could the results from this manuscript inform the development of such tools? Relatedly, how could the present findings inform the development of intervention programs for promoting positive PAP, parental responsiveness, and autonomy support?
RESPONSE: Thanks for the comment. We have actually added details considering the suggested quantitative measure development in accordance with answering to the earlier comment (page 13 lines 505-508). Considering the second comment, we think suggestions for these issues would be helpful for the reader. Thus, we decided to add suggestions considering these questions in the end of the conclusions (page 13 lines 523-525): “Consideration of the children’s perceptions of PAP could enhance quality of tailored PA intervention programs, e.g. through goal setting in PA counseling with parents focusing more on autonomy-supportive and less controlling PAP practices.”